# A Promising Thermodynamic Study of Hole Transport Materials to Develop Solar Cells: 1,3-Bis(*N*-carbazolyl)benzene and 1,4-Bis(diphenylamino)benzene

**DOI:** 10.3390/molecules27020381

**Published:** 2022-01-07

**Authors:** Juan Mentado-Morales, Arturo Ximello-Hernández, Javier Salinas-Luna, Vera L. S. Freitas, Maria D. M. C. Ribeiro da Silva

**Affiliations:** 1Instituto de Industrias, Universidad del Mar, Puerto Ángel, San Pedro Pochutla 70902, Oaxaca, Mexico; 2Procesos Bioalimentarios, Universidad Tecnológica de Tehuacán, Prolongación de la 1 Sur 1101, San Pablo Tepetzingo, Tehuacán 75859, Puebla, Mexico; 3Instituto de Ecología, Universidad del Mar, Puerto Ángel, San Pedro Pochutla 70902, Oaxaca, Mexico; jsl@angel.umar.mx; 4Centro de Investigação em Química da Universidade do Porto (CIQUP), Department of Chemistry and Biochemistry, Faculty of Science, University of Porto, Rua do Campo Alegre, P-4169-007 Porto, Portugal; vera.freitas@fc.up.pt (V.L.S.F.); mdsilva@fc.up.pt (M.D.M.C.R.d.S.)

**Keywords:** 1,3-bis(*N*-carbazolyl)benzene, 1,4-bis(diphenylamino)benzene, combustion energy, phase change enthalpies, heat capacities, thermogravimetry, spectra NMR

## Abstract

The thermochemical study of the 1,3-bis(*N*-carbazolyl)benzene (NCB) and 1,4-bis(diphenylamino)benzene (DAB) involved the combination of combustion calorimetric (CC) and thermogravimetric techniques. The molar heat capacities over the temperature range of (274.15 to 332.15) K, as well as the melting temperatures and enthalpies of fusion were measured for both compounds by differential scanning calorimetry (DSC). The standard molar enthalpies of formation in the crystalline phase were calculated from the values of combustion energy, which in turn were measured using a semi-micro combustion calorimeter. From the thermogravimetric analysis (TGA), the rate of mass loss as a function of the temperature was measured, which was then correlated with Langmuir’s equation to derive the vaporization enthalpies for both compounds. From the combination of experimental thermodynamic parameters, it was possible to derive the enthalpy of formation in the gaseous state of each of the title compounds. This parameter was also estimated from computational studies using the G3MP2B3 composite method. To prove the identity of the compounds, the ^1^H and ^13^C spectra were determined by nuclear magnetic resonance (NMR), and the Raman spectra of the study compounds of this work were obtained.

## 1. Introduction

Carbazole and its derivatives are a class of compounds commonly used in the synthesis of organic materials with application in the building of solar cells, being also found in natural products as alkaloids. The importance of the study of these compounds is due to the building block of their molecular structures, which offer many nuclear sites for the incorporation of different functional groups [1,2,3,4,5]. Similarly, triphenylamines and their derivatives have received attention since they have high solubility, thermal stability, and excellent optoelectronic properties [6,7,8]. In fact, it is due to these optoelectronic properties that these derivatives have an important role in the development of Hole Transport Materials (HTM) [9,10,11,12,13,14]. After a literature survey, as far as we know, there are no thermodynamic data for the two nitrogen-containing compounds proposed in this study, the 1,3-bis(*N*-carbazolyl) benzene (NCB) and 1,4-bis(diphenylamino)benzene (DAB) (Figure 1). These data are essential to provide information such as molecular packaging, cohesion energy related to the intermolecular interactions in the condensed phase, as well as an overview of the energy and their relationship with the molecular structure of this type of compound in the gaseous phase.

To reach the objectives proposed in this study—the determination of thermodynamic properties of NCB and DAB—essentially calorimetric and thermogravimetric techniques were used. The differential scanning calorimetry (DSC) enabled the assessment of compounds purity, as well as the measurement of the melting temperatures and enthalpies of fusion. The same technique was used to determine the heat capacities in the crystal phase over the temperature range of 274.15 to 332.15 K for both compounds. The enthalpies of vaporization of the two nitrogen compounds were measured by thermogravimetric analysis (TGA), being further combined with the enthalpy of fusion at reference temperature to calculate the enthalpy of sublimation. The combustion energies of the same compounds were obtained by combustion calorimetry, which was used to derive the corresponding enthalpy of formation in the crystalline phase. The gaseous state is the reference state to relate the energy of molecules to their structure and reactivity. To get there, the knowledge of the gas-phase enthalpy of formation, which does not contain energy terms for the intermolecular interactions, is crucial; therefore, this parameter was obtained indirectly from the combination of the enthalpy of formation in the crystalline phase and the enthalpy of sublimation at the reference temperature. The values of the gas-phase enthalpy of formation of these compounds were also derived from G3(MP2)//B3LYP calculations considering gas-phase hypothetical group substitution reactions for each compound.

## 2. Results and Discussion

### 2.1. Identification and DSC Parameters

The identification of NBC and DAB compounds was performed by nuclear magnetic resonance spectroscopy with respect to hydrogen-1 (^1^H NMR), carbon-13 (^13^C NMR), and Raman, using a Bruker Avance III 500 MHz and a fiber-coupled EZRAMAN-N of Enwave Optronics, respectively. The results on spectroscopic data as well as the ^1^H, ^13^C NMR, and RAMAN spectra of the NCB and DAB are given in the Appendix A.

Information regarding reagent suppliers, chemical data, and purity state of the two nitrogen compounds are given in Table 1. The final purities for NCB and DAB are (0.9996 ± 0.0003) and (0.9997 ± 0.0001), respectively, which were obtained by DSC.

The values of the enthalpy of fusion, $\Delta_{\mathrm{cr}}^{l}H_{m}^{o}\left(T_{\mathrm{fus}}\right)$, at the melting temperature $T_{\mathrm{fus}}$, were, respectively, (31.5 ± 0.1) kJ·mol^−1^ at (450.84 ± 0.08) K for NCB and (45.9 ± 0.2) kJ⋅mol^−1^ at (475.72 ± 0.04) K for DAB. The uncertainty given for the purity, enthalpy of fusion, and melting temperature corresponds to the standard deviation of the mean of a set of at least five measurements. The temperature and enthalpy of fusion of NCB are lower than for DAB, pointing to greater cohesion in the solid phase of DAB in comparison with NCB. The physical properties of the two nitrogen compounds together with other materials used throughout this work are shown in Appendix A.

The heat capacities in the crystalline phase, $C_{p}^{o}\left(\mathrm{cr}\right)$, at 298.15 K, for NCB and DAB, are, (445.16 ± 0.70) J⋅K^−1^⋅mol^−1^, and (458.94 ± 0.24) J⋅K^−1^⋅mol^−1^, respectively. The uncertainties of heat capacities at *T* = 298.15 K represent the standard deviation of the mean of at least four independent experiments series for each compound. The values of the heat capacities in the crystal phase for NCB and DAB, over the temperature range of (274.15 to 332.15) K, are given in Appendix A, respectively, of the Appendix A. The values of $C_{p}^{o}(\mathrm{cr})\;=f(T)$ were fitted over the temperature range mentioned by a second-degree polynomial function, expressed in Equations (1) and (2), for NCB and DAB, respectively, where the correlations coefficients for each fitted were greater than 0.9994 (Appendix A given in the Appendix A).
(1)$$C_{p,m}^{o}\left(\mathrm{cr},\;\mathrm{NCB}\right)/\left(J\cdot K^{-1}\cdot {\mathrm{mol}}^{-1}\right)=-0.0089{\left(T/K\right)}^{2}+6.948\left(T/K\right)-839.28\;r^{2}=0.9994$$
(2)$$C_{p,m}^{o}\left(\mathrm{cr},\;\mathrm{DAB}\right)/\left(J\cdot K^{-1}\cdot {\mathrm{mol}}^{-1}\right)=0.0012{\left(T/K\right)}^{2}+0.9022\left(T/K\right)+84.565\;r^{2}=0.9996$$

### 2.2. Combustion and Formation Enthalpy in Condensed Phase

Combustion experimental results obtained for NCB and DAB are shown in Appendix A, respectively, given in the Appendix A. The standard specific combustion energy, $\Delta_{c}u^{o}$, obtained for NCB is (−36,728.4 ± 2.1) J·g^−1^ and for DAB is (−37,814.2 ± 1.5) J·g^−1^; the quoted uncertainty corresponds to the standard deviation of the mean for seven experiments. These energies correspond to the idealized combustion reactions given by Equations (3) and (4), respectively.
(3)$$C_{30}H_{20}N_{2}\left(s\right)+{35O}_{2}\left(g\right)\to {30\mathrm{CO}}_{2}\left(g\right)+{10H}_{2}O\left(l\right)+N_{2}\left(g\right)$$
(4)$$C_{30}H_{24}N_{2}\left(s\right)+{36O}_{2}\left(g\right)\to {30\mathrm{CO}}_{2}\left(g\right)+{12H}_{2}O\left(l\right)+N_{2}\left(g\right)$$

The values of standard molar energies of combustion, $\Delta_{c}U_{m}^{o}$, standard molar enthalpies of combustion, $\Delta_{c}H_{m}^{o}$, and the standard molar enthalpies of formation, $\Delta_{f}H_{m}^{o}$, in crystalline phase, at *T* = 298.15 K, of the two nitrogen compounds are shown in Table 2. The standard molar enthalpy of formation of each compound was derived considering the corresponding idealized combustion reactions (Equations (3) or (4)) and the reported values at *T* = 298.15 K of $\Delta_{f}H_{m}^{o}\left(H_{2}O,\;l\right)$ = −(285.830 ± 0.042) kJ⋅mol^−1^ and $\Delta_{f}H_{m}^{o}\left({\mathrm{CO}}_{2},\;g\right)$ = −(393.51 ± 0.13) kJ⋅mol^−1^ [15].

### 2.3. Phase Changes by Thermogravimetry

Direct sublimation tests by thermogravimetry of the title compounds showed that there is no appreciable mass loss of the compounds before its melting temperature, therefore, its vaporization in the liquid phase was determined. The mass loss rate as a function of the temperature was obtained from the thermogravimetric measurements by using the TA Instruments Universal Analysis software. The vaporization enthalpies at average temperature, $\Delta_{l}^{g}H_{m}^{o}\left(T_{\mathrm{av}}\right)$, were obtained from the slope of the straight line obtained by plotting the dependence of $\ln\left(\left(dm/dt\right)\cdot T\;\right)$ versus 1/*T*, according to Equation (5).
(5)$$\ln\left(\frac{dm}{dt}\cdot T\right)=B-\frac{\Delta_{\mathrm{cr},l}^{g}H_{m}^{o}}{R}.\frac{1}{T}$$

Table 3 and Table 4 shows the representative mass loss, rate mass loss, and temperature ranges of (550.0 to 650.0) K and (500.0 to 600.0) K for NCB and DAB, respectively, as well as the fitted equations, the correlations coefficients, *r*^2^, and the uncertainties for the y-intercept σ_a_, and the slope σ_b_ of each experimental series. The vaporization enthalpy values are the weighted average, *μ*, which was calculated as $\mu =\sum_{i}^{N}\left(x_{i}/\sigma_{i}^{2}\right)/\sum_{i}^{N}\left(1/\sigma_{i}^{2}\right)$, whereas the uncertainties were calculated as $\sigma ={\left(N/\sum_{i}^{N}\left(1/\sigma_{i}^{2}\right)\right)}^{1/2}$, where *x*_i_ and *σ*_i_ are the experimental data of each *N* vaporization enthalpy and its respective uncertainty [18]. Figure 2 shows representative dependence of $\ln\left(\left(dm/dt\right)\cdot T\;\right)$ as a function of the temperature (1/*T*), derived from the vaporization experiments of each study compound. In this figure, a small decrease in the slopes is observed due to the wide range of the experimental work (100 K). However, *r*^2^ is close to unity, which indicates that the enthalpy of sublimation obtained in this work, remains constant and consequently this wide temperature range gives a greater margin in calculations where the enthalpy of sublimation of the NCB and DAB is concerned. Detailed data of all experiments are given in Appendix A.

The experimental values of vaporization enthalpies at average temperature, $\Delta_{l}^{g}H_{m}^{o}\left(T_{\mathrm{av}}\right)$, were adjusted to the melting temperature using Equation (6) suggested by Chickos et al. [19].
(6)$$\Delta_{l}^{g}H_{m}^{o}{(T}_{\mathrm{fus}})=\Delta_{l}^{g}H_{m}^{o}\left(T_{\mathrm{av}}\;\right)-[-\;0.0642{(T}_{\mathrm{av}}-T_{\mathrm{fus}})]$$

Subsequently, sublimation enthalpies were obtained from the addition of vaporization and fusion enthalpies (Equation (7)) and then corrected at *T* = 298.15 K with Equation (8) [19].
(7)$$\Delta_{\mathrm{cr}}^{g}H_{m}^{o}{(T}_{\mathrm{fus}})\;=\Delta_{\mathrm{cr}}^{l}H_{m}^{o}{(T}_{\mathrm{fus}})\;+\Delta_{l}^{g}H_{m}{(T}_{\mathrm{fus}})$$
(8)$$\Delta_{\mathrm{cr}}^{g}H_{m}^{o}(298.15\;K)\;=\Delta_{\mathrm{cr}}^{g}H_{m}^{o}{(T}_{\mathrm{fus}})-[-0{.032(T}_{\mathrm{fus}}-298.15\;K)]$$

Table 5 shows enthalpies of fusion, vaporization, and sublimation at experimental and reference temperatures of NCB and DAB, respectively. The uncertainties are given for the vaporization enthalpies at *T*_fus_, $\Delta_{l}^{g}H_{m}^{o}{(T}_{\mathrm{fus}})$, corresponds to the expanded uncertainties with a coverage factor of *k* = 1.96 for a confidence level of 0.95. The uncertainties in sublimation enthalpies at *T*_fus_, $\Delta_{\mathrm{cr}}^{g}H_{m}^{o}{(T}_{\mathrm{fus}})$, are the square root of the sum of the squared uncertainties of the vaporization and fusion enthalpies, both at *T*_fus_. The sublimation enthalpy of $\Delta_{\mathrm{cr}}^{g}H_{m}^{o}\left(298.15\;K\right)$ = (164.5 ± 1.5) kJ⋅mol^−^^1^ and $\Delta_{\mathrm{cr}}^{g}H_{m}^{o}(298.15\;K)$ = (163.6 ± 2.3) kJ⋅mol^−^^1^ for NCB and DAB, respectively, shown that there is a difference between these values of 0.9 kJ⋅mol^−^^1^, which indicates that the intermolecular interactions in the crystalline phase are very similar.

### 2.4. Experimental Enthalpy of Formation in Gas Phase

The standard molar formation enthalpies in the gas phase, at *T* = 298.15 K, $\Delta_{f}H_{m}^{o}(g,\;298.15\;K)$, for NCB and DAB were obtained by the combination of the values of the enthalpy of sublimation and the enthalpy of formation in the crystalline phase, whose values are presented in Table 6. The uncertainties in the gas phase were calculated through the root sum square method.

### 2.5. Computational Gas-Phase Enthalpy of Formation

The optimized molecular geometries obtained with B3LYP/6-31G(*d*) level of theory for NCB and DAB molecules, together with electrostatic potential energy map (EPEM), are given in Figure 3 and Figure 4, respectively. These ones correspond to the global minimum conformer on the potential energy surface. As can be seen, the NCB molecule has a plane of symmetry that passes through carbons 2 and 5 of the benzene ring. The carbazole moieties are bonded by the nitrogen atom to the benzene ring at 1 and 3 positions, being that both carbazoles are bended relatively to the benzene ring, resembling a “butterfly” shape. Both planes containing the carbazole moieties are in symmetrical positions (picture of each other), and these planes are offset by 56° relative to the plane of the benzene ring.

In the DAB molecule, both diphenyl moieties are bonded by the nitrogen atom to the benzene ring at 2 and 4 positions. This molecule has a plane of symmetry that passes through the two nitrogen atoms. In a previous computational study performed by some of us, focusing on the energetic and reactivity properties of diphenylamine [22], it was verified that the diphenylamine equilibrium geometry is influenced by two effects: the steric repulsion between the closest hydrogens of neighbouring rings and the conjugation of the electronic π systems, being the twisted conformation, out of three possible conformations, the minimum on the potential energy surface. In the same study, it was demonstrated that for the geometry optimization using the B3LYP/6-31G(*d*) method, the torsion angle of the benzene rings in the diphenylamine obtained was 22°. Also, in the DAB molecule, it is verified that the diphenylamine moieties acquire the twisted form, and for the same DFT method, the torsion angles of the benzene rings are higher: 40°.

In both Figure 3 and Figure 4 are also shown the graphical representations of the electrostatic potential energy maps (EPEM) for NCB and DAB obtained from the Natural Bond Orbital (NBO) calculations, performed using the NBO 3.1 program [23] as implemented in the Gaussian 03 package [24], at the B3LYP/6-31G(*d,p*) level of theory [25,26]. The conventional choice for the color map is the visible spectrum to represent the varying intensities of the electrostatic potential energy values, i.e., red for the lowest electrostatic potential energy value (higher charge density), blue for the highest electrostatic potential energy value (lower charge density), and green for the electrostatic potential energy values near zero. There is no definitive criterion for choosing the isodensity value. Though arbitrary, there is a broad acceptance of a standard density of 0.002 e⋅*a*_0_^−3^, which is thought to best represent the size and shape of a molecule because it corresponds to the van der Waals atomic radii [27]. These graphical representations are useful for visualizing regions of the molecule where the charge is concentrated (red regions) or depleted (blue regions). Observing the EPEMs obtained for the molecules under study, the number of rings noticeably influences the electronic distribution, verifying that in the case of NCB there is a higher electron density in the carbazole rings, while in the case of DAB, the fact that benzene rings of diphenylamine are in twist positions, consequently having an annular break, there is a lower electron density in the diphenyl rings.

The hypothetical gas-phase substitution group reactions developed for NCB and DAB together with the corresponding gas-phase enthalpy of formation of the NCB and DAB are reported in Table 7 and Table 8, respectively.

For NCB are presented four hypothetical gas-phase substitution reactions with values for the enthalpy of formation between (529 to 549) kJ⋅mol^−1^, having a mean value of (539 ± 26) kJ⋅mol^−1^. In the case of the computational study of DAB, five hypothetical gas-phase substitution reactions were used with values for the enthalpy of formation between (566 to 575) kJ⋅mol^−1^, having a mean value of (570.7 ± 8.3) kJ⋅mol^−1^.

### 2.6. Comparison and Comments on the Gas-Phase Enthalpy of Formation of NCB and DAB

The values of the experimental and computational gas-phase enthalpies of the formation of the two nitrogen compounds are not in good agreement. The observed dissimilarity between the experimental and computational results on the formation enthalpy of the gaseous compounds may be justified by some limitations on the application of the selected computational method for this kind of molecule. In fact, the composite method G3(MP4)//B3LYP is one of the most used theoretical methods for large molecules due to the shorter calculation time needed, although considering the huge molecular weight of these molecules, some restrictions may support the difference. We feel that for the computational study of this kind of structure it may be convenient to explore the computational methodologies in order to improve their use to these systems. Currently, we found some limitations on the availability of reliable experimental information to validate new approaches.

## 3. Materials and Methods

### 3.1. Compounds and Differential Scanning Calorimetry (DSC)

The compounds studied in this work were supplied by Sigma-Aldrich (Merck KGaA, Darmstadt, Germany), with purities of the order of 0.97 for 1,3-bis(*N*-carbazolyl)benzene (NCB) and for 1,4-bis(diphenylamino)benzene (DAB). The purity required to perform combustion calorimetry experiments is higher than 0.99. Due to the low purity of the compounds, both were purified by recrystallization using a 60% ethanol and 40% ethyl acetate mixture. After the purification process, the compounds were analysed by nuclear magnetic resonance spectroscopy with respect to hydrogen-1 (^1^H NMR) and carbon-13 (^13^C NMR). Additionally, the purity was assessed by differential scanning calorimetry (DSC) using the fractional fusion technique [28].

The DSC Q2000 calorimeter (TA Instruments, New Castle, DE, USA) was used to perform the purity control, and the determination of the melting temperatures, and fusion enthalpies, which was calibrated with the fusion of high-purity metallic indium. In all experiments, masses of, approximately, 3 mg were placed in a non-hermetic aluminium crucible and then heated from (433 to 463) K for NCB and from (453 to 493) K for DAB, and a scanning rate of 5 K min^−1^ and flow rate of 50 mL min^−1^ of nitrogen were applied.

In addition, the heat capacity of the compounds was determined with a DSC 8000 calorimeter (PerkinElmer Inc., MA, USA) over the temperature range of (274.15 to 332.15) K using synthetic sapphire as a reference and using the two steps method [29,30].

The masses of all compounds were measured in a balance UMX2 (Mettler Toledo, Ohio, USA) with a resolution of ±0.1 µg.

### 3.2. Combustion Calorimetry

The specific combustion energies in the crystalline phase for NCB and DAB were obtained by combustion calorimetry using a semi-micro oxygen bomb Parr 1109A (Parr Instrument Company, IL, USA), which has an internal volume of 0.022 dm^3^ [31]. The calorimeter was recently calibrated with benzoic acid (SRM 39j) following the procedure suggested by Coops et al. [32]. The energy equivalent of the device obtained was, $\varepsilon \left(\mathrm{calor}\right)$ = (2042.4 ± 0.3) J⋅K^−1^, where the uncertainty quoted corresponds to the standard deviation of the mean [33].

The test combustion experiments for NCB and DAB showed that there is not complete oxidation because residual carbon traces were observed, therefore, paraffin oil was used such as auxiliary combustion material, which has a specific combustion energy of $\Delta_{c}u^{o}$ = −(46,239.8 ± 6.5) J⋅g^−1^ [34]. The samples and materials used were placed in a platinum crucible and 0.0001 dm^3^ of deionized water was added to ensure the thermodynamic equilibrium at the end of the combustion process. The bomb was purged five times with oxygen of high purity (*x* = 0.99999) and then filled to a pressure of 3.04 MPa. The water of the isothermal jacket of the calorimeter system was maintained and regulated at a constant temperature of 298.15 K by a refrigerated circulator Polyscience 9502 (PolyScience, IL, USA). The masses of the samples, cotton thread, auxiliary material, crucible and platinum wire were weighed on a DV215CD balance (Ohaus, Parsippany, NJ, USA) (accuracy ±0.01 mg). The electrical energy for the ignition of the sample was supplied with a 2901 Parr ignition unity, which provides 4.184 J to a platinum fuse wire attached to the sample by a cotton thread. The cotton thread used has a massic energy of combustion of $\Delta_{c}u^{o}$ = −(16,945.2 ± 4.2) J⋅g^−1^, and an empirical formula of C_1_H_1.742_O_0.901_ [35]. The temperature acquisition was generated using a 5642 Hart Scientific thermistor (diameter = 3.18 mm, length = 229 mm, and resistance = 4 kΩ), which is coupled to a Keithley 2010 digital multimeter (Final Test S.A. de C. V, Tijuana B.C., México) (sensitivity: 10^−6^ kΩ) plugged on a personal computer for automatic data collection. The corrected temperature rise for each experiment of combustion was calculated by the Regnault–Pfaundler method [36]. At the end of the combustion experiments, no carbon residue from incomplete combustion was observed since combustion aid was used, and the nitric acid formed was quantified by acid–base volumetric titrations [37,38]. The correction for the formation of nitric acid in the calculation of the combustion energy of each sample was based on the value of −59.7 kJ⋅mol^−1^ for the formation of 0.1 mol⋅dm^−3^ of HNO_3_(l) from N_2_(g), O_2_(g), and H_2_O(l) [39,40]. The specific energy as a function of pressure, ${\left(\partial u/\partial p\right)}_{T}$, at *T* = 298.15 K for the study compounds was assumed to be −0.2 J∙g^−1^∙MPa^−1^, a typical value for organic compounds [41]. Washburn corrections were applied as described by Hubbard et al. [42]. The relative atomic masses used in this work followed the recommendations of the 2013 IUPAC Commission [43].

### 3.3. Thermogravimetry

The principles to the determination of the enthalpies of vaporization/sublimation by using a thermogravimetric device are based on the kinetic theory of gases, in which the process of evaporation of a substance is expressed with the Langmuir equation (Equation (9)) [44,45,46,47].
(9)$$\left(dm/dt\right)\left(1/A\right)=p\alpha \;\sqrt{M/2\pi RT}$$

In Equation (9), d*m*/d*t* is the mass loss rate, *p* is the vapour pressure at temperature *T*, *A* is the exposed area of the sample in the vaporization/sublimation process, *M* is the molar mass, *R* is the universal gas constant and *α* is the vaporization coefficient [48,49]. By combining the Clausius–Clapeyron equation with the Langmuir equation, it is possible to calculate the standard molar enthalpy of vaporization/sublimation, using Equation (5) given above. In the Equation (5), B=ln(DMS/R)+C, where the diffusion phenomena were considered [50,51]. This equation is used to realize both, isothermal and dynamic experiments and it is not necessary to determine the vapour pressure directly, only a thermogravimetric device is needed to measure with accuracy and precision the mass loss rate as a function of the temperature of the study compounds. The enthalpies of the phase transition are accurately obtained by fitting the reciprocal of the temperature, 1/*T*, against $\ln\left((dm/dt)\cdot T\right)$. The Langmuir equation has been successfully applied for measurements of vaporization/sublimation of numerous organic compounds [52,53,54,55,56,57,58,59,60].

Measurement of the rate of mass loss as a function of temperature with a TGA Q500 (TA Instruments, New Castle, DE, USA) was performed. This equipment has a thermobalance with a temperature and mass sensitivities of ±0.1 K and ±0.1 μg, respectively, and a maximum capacity of 1 g. In the thermogravimetric experiments, samples of 16.6 and 10.5 mg for NCB and DAB, respectively, were used. The initial and final temperatures of vaporization experiments were (550 to 650) K and (500 to 600) K, respectively, and a heating rate of 10.0 K⋅min^−1^ with a flow rate of nitrogen of 100 cm^3^⋅min^−1^, were applied.

In order, to verify that no decomposition process occurred during the thermogravi-metric experiments. The substance was recovered at the exit of the sublimation oven and two tests were carried out. (1) ^1^H, ^13^C spectra were obtained by NMR, for the case of the two samples, the presence of spectral signals different from those obtained and reported in the Appendix A) was not observed. (2) The recovered substance was analysed by DSC and in both cases, for each compound (NCB and DAB) there was no difference in the melting thermograms. Details about the device, calibrations, and testing were recently reported [51].

### 3.4. Computational Method

A state-of-the-art computational thermochemistry study was performed to support the experimental values. The energies of the molecules were calculated using the G3(MP2)//B3LYP composite method included in the Gaussian 09 suite of programs [24,61]. The gas-phase enthalpies of formation of the NCB and DAB compounds were calculated using hypothetical gas-phase substitution group reactions (namely, isodesmic and homodesmotic reactions). This kind of reaction involves species with structural analogies to those of the compounds under study, where the reactants and the products of the reactions are constituted by molecules with related structural and formal bond types, in order to minimize differential errors in the electronic structure computations, enabling the increase in the accuracy of the results. In previous works carried out in the scope of thermochemistry of heteropolycyclic compounds carried out by some of us, this type of equation showed results in agreement with the experimental values [62,63,64]. The drawback of these reactions is the requirement of accurate experimental values of the gas-phase enthalpies of formation of each molecule involved and that may not always be an easy task.

The gas-phase standard molar enthalpy of each hypothetical reaction was calculated taking into account Equations (10) and (11). The enthalpy of reaction, $\Delta_{R}H_{m}^{{}^{\circ}}$, at *T* = 298.15 K, was obtained computationally from the absolute standard enthalpies, $H_{298.15K}^{{}^{\circ}}$, of each species. The rearrangement of Equation (11) and the knowledge of the experimental standard molar gas-phase enthalpies of formation of all the auxiliary species used enabled the calculation of the species under study. The G3(MP2)//B3LYP absolute enthalpies, $H_{298.15K}^{{}^{\circ}}$, and the experimental enthalpies of formation in the gas phase, $\Delta_{f}H_{m}^{{}^{\circ}}\left(g\right)$, of the molecular species used, are given in Appendix A in the Appendix A.
(10)$$\Delta_{R}H_{m}^{{}^{\circ}}=\sum H_{298.15K}^{{}^{\circ}}\left(\mathrm{products}\right)-\sum H_{298.15K}^{{}^{\circ}}\left(\mathrm{reagents}\right)$$
(11)$$\Delta_{R}H_{m}^{{}^{\circ}}=\sum \Delta_{f}H_{m}^{{}^{\circ}}\left(\mathrm{products}\right)-\sum \Delta_{f}H_{m}^{{}^{\circ}}\left(\mathrm{reagents}\right)$$

## 4. Conclusions

Experimental thermochemical properties with high accuracy for NCB and DAB were obtained using DSC, combustion calorimetry, and thermogravimetry. Moreover, theoretical enthalpies of formation in the gas phase were obtained. The results of the standard molar enthalpies of formation in the gas phase of NCB and DAB, respectively, showed that the difference in energy between both compounds is about 27.8 kJ⋅mol^−1^, and in this way, we can say the NCB is enthalpically more stable than DAB.

## Figures and Tables

**Figure 1 molecules-27-00381-f001:**
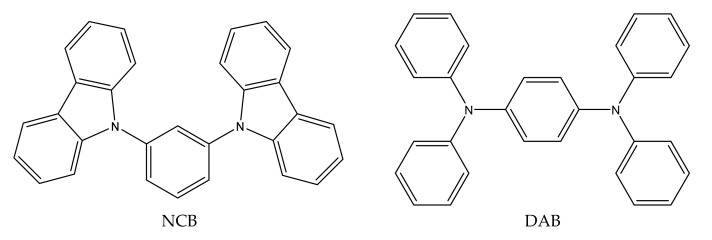
Molecular structures of 1,3-bis(N-carbazolyl)benzene (NCB) and 1,4-bis(diphenylamino)benzene (DAB) compounds.

**Figure 2 molecules-27-00381-f002:**
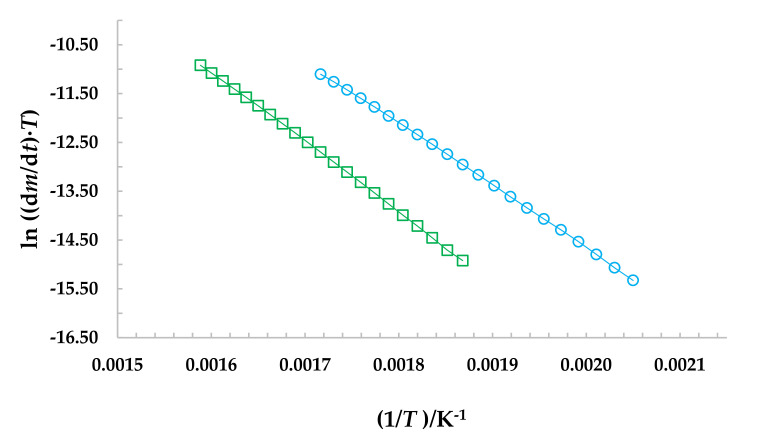
Representative dependence of $\ln\left(\left(dm/dt\right)\cdot T\;\right)$ versus 1/*T* of each compound. **□** 1,3-bis(*N*-carbazolyl)benzene (NCB); ○ 1,4-bis(diphenylamino)benzene (DAB).

**Figure 3 molecules-27-00381-f003:**
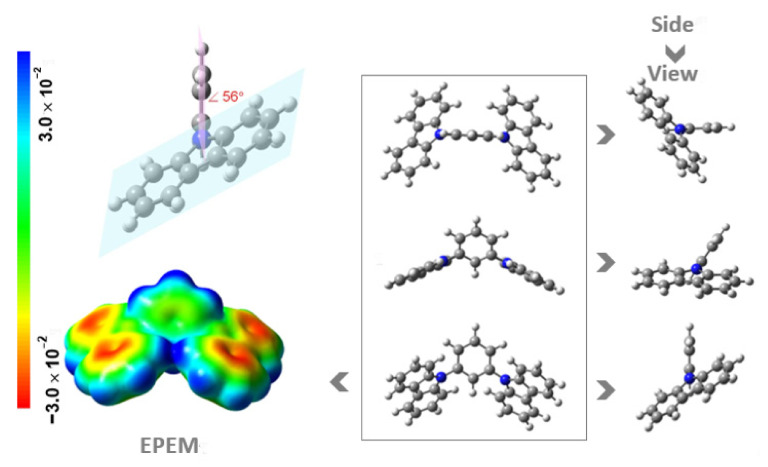
Different views of molecular structures and electrostatic potential energy maps (EPEM) mapped onto the surface of the molecule, with an isodensity surface value of 0.002 *e*⋅*a*_0_^−3^ (where *a*_0_ is the Bohr radius; 1 *e* corresponds to 1.6021766208 × 10^−19^ C [20]; 1 *a*_0_ corresponds to 5.2917721067 × 10^−11^ m [21]), calculated for NCB. The dihedral angle (in red) was calculated based on planes defined by each benzene ring and the carbazole units. Colour code for spheres: grey, C; blue, N; white H.

**Figure 4 molecules-27-00381-f004:**
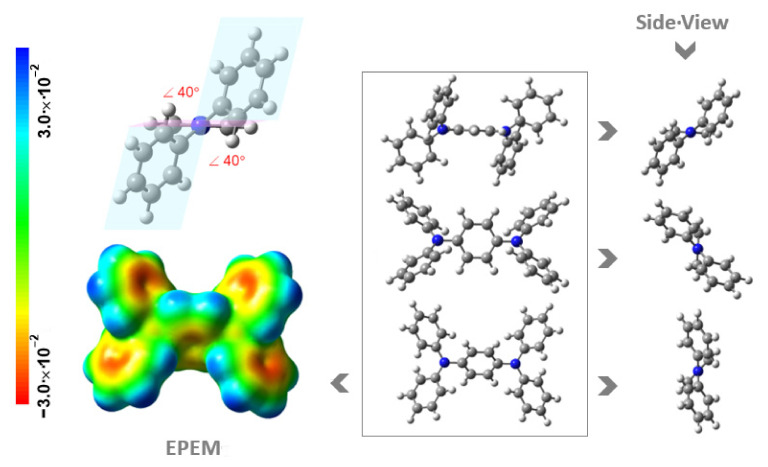
Different views of molecular structures and the electrostatic potential energy maps (EPEM) mapped onto the surface of the molecule, with an isodensity surface value of 0.002 *e*⋅*a*_0_^−3^ (where *a*_0_ is the Bohr radius; 1 *e* corresponds to 1.6021766208 × 10^−19^ C [20]; 1 *a*_0_ corresponds to 5.2917721067 × 10^−11^ m [21]), calculated for DAB. The dihedral angle (in red) was calculated based on planes defined by each benzene ring and the benzene rings of the diphenyl amine moiety. Colour code for spheres: grey, C; blue, N; white, H.

**Table 1 molecules-27-00381-t001:** Chemical data, source, and purities of the compounds used in this work.

Commercial Name (Acronym)	CAS Number	Source	Initial Mass Fraction Purity ^1^	Purification Method ^2^	Final Mass Fraction Purity ^3^
1,3-bis(N-carbazolyl)benzene (NCB)	550378-78-4	Sigma Aldrich^®^	0.97	recrystallization	0.9996 ± 0.0003
1,4-bis(diphenylamino)benzene (DAB)	14118-16-2		0.97	recrystallization	0.9997 ± 0.0001

^1^ Value obtained in the supplier´s analysis certificate. ^2^ Recrystallization realized with 60% ethanol and 40% ethyl acetate mixture. ^3^ Differential scanning calorimetry (DSC Q2000, TA Instruments, New Castle, DE, USA) was the analyses method used; the uncertainties of fraction purity represent the standard deviation of the mean.

**Table 2 molecules-27-00381-t002:** Derived standard (*p*° = 0.1 MPa) molar values of compounds in the crystalline phase, at *T* = 298.15 K.

Compound	$-\Delta_{c}U_{m}^{\;o}\left(cr\right)/kJ\cdot {mol}^{-1}$	$-\Delta_{c}H_{m}^{\;o}\left(cr\right)/kJ\cdot {mol}^{-1}$	$-\Delta_{f}H_{m}^{\;o}\left(cr\right)/kJ\cdot {mol}^{-1}$
NCB	15,003.3 ± 6.6 ^1^	15,013.2 ± 6.6 ^1^	349.6 ± 7.7 ^2^
DAB	15,599.3 ± 6.7 ^1^	15,611.7 ± 6.7 ^1^	376.4 ± 7.8 ^2^

^1^ The uncertainty corresponds to the expanded uncertainty determined from the combined standard uncertainty (which include the contribution of the calibration with benzoic acid and the use of paraffin as auxiliary aid) and the coverage factor k = 2 (for a 0.95 level of confidence). ^2^ The uncertainty corresponds to the expanded uncertainty determined from the combined standard uncertainty (which include the contribution of the species involved in the combustion reactions presented in Equations (3) and (4)) and the coverage factor *k* = 2 (for a 0.95 level of confidence) [16,17].

**Table 3 molecules-27-00381-t003:** Representative experimental data for the determination of the vaporization enthalpy of 1,3-bis(*N*-carbazolyl)benzene (NCB) in the temperature range 550.0 to 650.0 K ^1^.

T/K	m/mg	$(dm⁄dt)\cdot 10^{9}/\mathrm{kg}\cdot s^{-1}$	$\left(1/T\right)\cdot 10^{3}{/K}^{-1}$	$-\ln\left[\left(dm/dt\right)\cdot T\right]$
550.0	16.6151	0.6021	1.818	14.921
555.0	16.5950	0.7384	1.802	14.708
560.0	16.5695	0.9424	1.786	14.455
565.0	16.5376	1.1899	1.770	14.213
570.0	16.4976	1.4724	1.754	13.991
575.0	16.4478	1.8404	1.739	13.759
580.0	16.3861	2.2805	1.724	13.536
585.0	16.3098	2.8146	1.709	13.317
590.0	16.2163	3.4351	1.695	13.109
595.0	16.1018	4.1887	1.681	12.902
600.0	15.9623	5.0904	1.667	12.699
605.0	15.7934	6.1669	1.653	12.499
610.0	15.5897	7.4318	1.639	12.304
615.0	15.3450	8.8927	1.626	12.116
620.0	15.0535	10.6169	1.613	11.931
625.0	14.7053	12.6390	1.600	11.749
630.0	14.2936	14.8808	1.587	11.577
635.0	13.8080	17.4926	1.575	11.408
640.0	13.2393	20.4814	1.563	11.242
645.0	12.5737	23.8926	1.550	11.080
650.0	11.7970	27.8601	1.538	10.919
$\mathrm{Series}\;1.\;\ln\left(\left(dm/dt\right)\cdot T\;\right)$ = 11.3–14,396.9/*T*; *r*^2^ = 0.9997; σ_a_ = 0.10; σ_b_ = 58.5; $\;\Delta_{l}^{g}H_{m}^{o}$(600.0 K)/kJ⋅mol^−1^ = 119.7 ± 0.5				
$\mathrm{Series}\;2.\;\ln\left(\left(dm/dt\right)\cdot T\;\right)$ = 10.9–14,198.1/*T*; *r*^2^ = 0.9998; σ_a_ = 0.07; σ_b_ = 42.2; $\;\Delta_{l}^{g}H_{m}^{o}$(600.0 K)/kJ⋅mol^−1^ = 118.0 ± 0.4				
$\mathrm{Series}\;3.\;\ln\left(\left(dm/dt\right)\cdot T\;\right)$ = 11.1–14,281.6/*T*; *r*^2^ = 0.9997; σ_a_ = 0.10; σ_b_ = 59.3; $\;\Delta_{l}^{g}H_{m}^{o}$(600.0 K)/kJ⋅mol^−1^ = 118.7 ± 0.5				
$\mathrm{Series}\;4.\;\ln\left(\left(dm/dt\right)\cdot T\;\right)$ = 11.1–14,240.9/*T*; *r*^2^ = 0.9997; σ_a_ = 0.10; σ_b_ = 59.2; $\;\Delta_{l}^{g}H_{m}^{o}$(600.0 K)/kJ⋅mol^−1^ = 118.4 ± 0.5				
$\mathrm{Weighted}\;\mathrm{average}\;\mathrm{value}:\;<\Delta_{l}^{g}H_{m}^{o}$(NCB, 600 K)>/kJ⋅mol^−1^ = 118.6 ± 0.4				

^1^ Parameters σ_a_ and σ_b_ represents the standard deviation of the intercept and slope of the function $\ln\left(\left(dm/dt\right)\cdot T\;\right)\;$ vs. 1/*T*. The uncertainty for each vaporization enthalpy value was computed as σ_b_⋅*R·*10^–3^. The weighted average value *μ* and its standard deviation σ, were calculated as $\mu =\sum_{i}^{N}\left(x_{i}/\sigma_{i}^{2}\right)/\sum_{i}^{N}\left(1/\sigma_{i}^{2}\right)$ and $\sigma ={\left(N/\sum_{i}^{N}\left(1/\sigma_{i}^{2}\right)\right)}^{1/2}$ where *x*_i_ and *σ*_i_ are the experimental data of each *N* vaporization enthalpy and its respective uncertainty [18].

**Table 4 molecules-27-00381-t004:** Representative experimental data for the determination of the vaporization enthalpy of 1,4-Bis(diphenylamino)benzene (DAB) in the temperature range 500.0 to 600.0 K ^1^.

T/K	m/mg	$(dm⁄dt)\cdot 10^{9}/\mathrm{kg}\cdot s^{-1}$	$\left(1/T\right)\cdot 10^{3}{/K}^{-1}$	$-\ln\left[\left(dm/dt\right)\cdot T\right]$
500.00	10.6641	0.4411	2.000	15.327
505.00	10.6492	0.5658	1.980	15.068
510.00	10.6296	0.7364	1.961	14.795
515.00	10.6044	0.9464	1.942	14.534
520.00	10.5724	1.1944	1.923	14.292
525.00	10.5325	1.4796	1.905	14.068
530.00	10.4828	1.8358	1.887	13.843
535.00	10.4210	2.2892	1.869	13.613
540.00	10.3441	2.8443	1.852	13.386
545.00	10.2492	3.5138	1.835	13.166
550.00	10.1321	4.2965	1.818	12.956
555.00	9.9888	5.2672	1.802	12.743
560.00	9.8140	6.4049	1.786	12.538
565.00	9.6021	7.7353	1.770	12.341
570.00	9.3460	9.3120	1.754	12.146
575.00	9.0390	11.1517	1.739	11.957
580.00	8.6720	13.2946	1.724	11.773
585.00	8.2371	15.7318	1.709	11.596
590.00	7.7230	18.5333	1.695	11.424
595.00	7.1203	21.6737	1.681	11.259
600.00	6.4172	25.1040	1.667	11.103
$\mathrm{Series}\;1.\;\ln\left(\left(dm/dt\right)\cdot T\;\right)$ = 10.1–12,705.0/*T*; *r*^2^ = 0.9997; σ_a_ = 0.09; σ_b_ = 51.7; $\;\Delta_{l}^{g}H_{m}^{o}$(550.0 K)/kJ⋅mol^−1^ = 105.6 ± 0.4				
$\mathrm{Series}\;2.\;\ln\left(\left(dm/dt\right)\cdot T\;\right)$ = 10.7–13,059.5/*T*; *r*^2^ = 0.9993; σ_a_ = 0.15; σ_b_ = 80.0; $\;\Delta_{l}^{g}H_{m}^{o}$(550.0 K)/kJ⋅mol^−1^ = 108.6 ± 0.7				
$\mathrm{Series}\;3.\;\ln\left(\left(dm/dt\right)\cdot T\;\right)$ = 10.7–13,042.7/*T*; *r*^2^ = 0.9993; σ_a_ = 0.14; σ_b_ = 77.1; $\;\Delta_{l}^{g}H_{m}^{o}$(550.0 K)/kJ⋅mol^−1^ = 108.4 ± 0.6				
$\mathrm{Series}\;4.\;\ln\left(\left(dm/dt\right)\cdot T\;\right)$ = 10.8–13,049.3/*T*; *r*^2^ = 0.9992; σ_a_ = 0.15; σ_b_ = 84.3; $\;\Delta_{l}^{g}H_{m}^{o}$(550.0 K)/kJ⋅mol^−1^ = 108.5 ± 0.7				
$\mathrm{Weighted}\;\mathrm{average}\;\mathrm{value}:\;<\Delta_{l}^{g}H_{m}^{o}$(DAB, 550 K)>/kJ⋅mol^−1^ = 107.2 ± 0.6				

^1^ Parameters σ_a_ and σ_b_ represent the standard deviation of the intercept and slope of the function $\ln\left(\left(dm/dt\right)\cdot T\;\right)\;$ vs. 1/*T*. The uncertainty for each vaporization enthalpy value was computed as *σ*_b_⋅*R·*10^−3^. The weighted average value *μ* and its standard deviation σ, were calculated as $\mu =\sum_{i}^{N}\left(x_{i}/\sigma_{i}^{2}\right)/\sum_{i}^{N}\left(1/\sigma_{i}^{2}\right)$ and $\sigma ={\left(N/\sum_{i}^{N}\left(1/\sigma_{i}^{2}\right)\right)}^{1/2}$, where *x*_i_ and *σ*_i_ are the experimental data of each *N* vaporization enthalpy and its respective uncertainty [18].

**Table 5 molecules-27-00381-t005:** Values of fusion and vaporization enthalpies obtained at the experimental temperature for the two nitrogen compounds, and vaporization and sublimation enthalpies at the melting temperature and sublimation enthalpies at *T* = 298.15 K and *p*° = 0.1 MPa.

Compound	$T_{\mathrm{fus}}/K$	$T_{\mathrm{av}}/K$	$\frac{\Delta_{\mathrm{cr}}^{l}H_{m}^{o}{(T}_{\mathrm{fus}})}{\mathrm{kJ}\cdot {\mathrm{mol}}^{-1}}$	$\frac{\Delta_{l}^{g}H_{m}^{o}{\;(T}_{\mathrm{av}})}{\mathrm{kJ}\cdot {\mathrm{mol}}^{-1}}$	$\frac{\Delta_{l}^{g}H_{m}^{o}{(T}_{\mathrm{fus}})}{\mathrm{kJ}\cdot {\mathrm{mol}}^{-1}}$	$\frac{\Delta_{\mathrm{cr}}^{g}H_{m}^{o}{(T}_{\mathrm{fus}})}{\mathrm{kJ}\cdot {\mathrm{mol}}^{-1}}$	$\frac{\Delta_{\mathrm{cr}}^{g}H_{m}^{o}(298.15\;K)}{\mathrm{kJ}\cdot {\mathrm{mol}}^{-1}}$
NCB	450.84 ± 0.08 ^1,2^	600.0	31.5 ± 0.1 ^1,2^	118.6 ± 0.4 ^3^	128.2 ± 0.8 ^4,6^	159.6 ± 0.8 ^5,7^	164.5 ± 1.5 ^4,8^
DAB	475.72 ± 0.04 ^1,2^	550.0	45.9 ± 0.2 ^1,2^	107.2 ± 0.6 ^3^	112.0 ± 1.2 ^4,6^	157.9 ± 1.2 ^5,7^	163.6 ± 2.3 ^4,8^

^1^ Experimental value obtained by DSC. ^2^ Standard deviation of the mean. ^3^ Experimental values obtained by thermogravimetry. ^4^ Expanded uncertainty with a coverage factor of k = 1.96 to ensure a 0.95 confidence level. ^5^ Uncertainty obtained through the root sum square method. ^6^ Value obtained with Equation (6). ^7^ Value obtained with Equation (7). ^8^ Value obtained with Equation (8).

**Table 6 molecules-27-00381-t006:** Standard (*p*° = 0.1 MPa) molar enthalpies of sublimation and formation in gas-phase at *T* = 298.15 K for NCB and DAB, respectively.

Compound	$\frac{\Delta_{\mathrm{cr}}^{g}H_{m}^{o}({298.15\;K)\;}^{a}}{\mathrm{kJ}\cdot {\mathrm{mol}}^{-1}}$	$\frac{\Delta_{f}H_{m}^{o}(\mathrm{cr},\;{298.15\;K)\;}^{b}}{\mathrm{kJ}\cdot {\mathrm{mol}}^{-1}}$	$\frac{\Delta_{f}H_{m}^{o}{\left(g,\;298.15\;K\right)}^{\;c}}{\mathrm{kJ}\cdot {\mathrm{mol}}^{-1}}$
NCB	164.5 ± 1.5	349.6 ± 7.7	514.1 ± 7.8
DAB	163.6 ± 2.3	376.4 ± 7.8	540.0 ± 8.1

*^a^* Expanded uncertainty with a coverage factor *k* = 1.96 to ensure a 0.95 confidence level. *^b^* The uncertainty corresponds to the expanded uncertainty determined from the combined standard uncertainty (which include the contribution of the species involved in the combustion reactions presented in Equations (3) and (4)) and the coverage factor k = 2 (for a 0.95 level of confidence). *^c^* Uncertainty calculated through the root sum square method.

**Table 7 molecules-27-00381-t007:** Hypothetical gas-phase reactions and corresponding enthalpies of reactions, $\Delta_{r}H_{m}^{\text{°}}$ (g), together with the calculated gas-phase enthalpy of formation, $\Delta_{f}H_{m}^{\text{°}}\left(g\right)$, of NCB, at *T* = 298.15 K.

	Hypothetical Gas-Phase Reactions	ΔrHm°kJ·mol−1	ΔfHm°kJ·mol−1
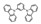	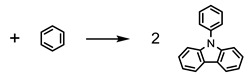	(R7.1)	0.95	548.65
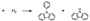	(R7.2)	−20.31	541.41
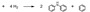	(R7.3)	−11.50	529.50
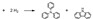	(R7.4)	−2.54	534.54
		Mean value:	539 ± 26 ^1^

^1^ The quoted uncertainty defines an interval having a 0.95 level of confidence (coverage factor used k = 3.182 for 3 degrees of freedom).

**Table 8 molecules-27-00381-t008:** Hypothetical gas-phase reactions and corresponding enthalpies of reactions, $\Delta_{r}H_{m}^{\text{°}}$ (g), together with the calculated gas-phase enthalpy of formation, $\Delta_{f}H_{m}^{\text{°}}\left(g\right)$, of DAB, at *T* = 298.15 K.

	Hypothetical Gas-Phase Reactions	ΔrHm°kJ·mol−1	ΔfHm°kJ·mol−1
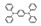	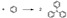	(R8.1)	−0.29	571.69
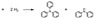	(R8.2)	−24.29	568.99
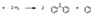	(R8.3)	−48.29	566.29
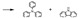	(R8.4)	−39.32	571.32
	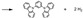	(R8.5)	−36.78	575.28
		Mean value:	570.7 ± 8.3 ^1^

^1^ The quoted uncertainty defines an interval having a 0.95 level of confidence (coverage factor used k = 2.776 for 4 degrees of freedom).

## Data Availability

Not applicable.

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
