# Peer review of "A Promising Thermodynamic Study of Hole Transport Materials to Develop Solar Cells: 1,3-Bis(N-carbazolyl)benzene and 1,4-Bis(diphenylamino)benzene"

_molecules, 2022, doi:10.3390/molecules27020381_

Round 1

Reviewer 1 Report

The present manuscript deals with the thermochemical study of the 1,3-Bis(N-carbazolyl)benzene (NCB) and 1,4-Bis(diphenylamino)benzene (DAB) using both the combustion calorimetric (CC) and thermogravimetric techniques. The molar heat capacities over the temperature range of (274.15 to 332.15) K, as well as the melting temperatures and enthalpies of fusion were measured for both compounds by differential scanning calorimetry (DSC). The standard molar enthalpies of formation in the crystalline phase were calculated from the values of combustion energy (measured by a semi-micro combustion calorimeter). This study represents a continuation of a research theme that involves two of the authors (M.D.M.D.R..S. and V.L.S.F.) since many years: the influence of the structure on the energy and reactivity of  many molecular substances with particular reference to nitrogen heterocyclic compounds.   The following sections are clearly presented and well-organized: Combustion and Formation Enthalpy in Condensed Phase Computational gas-phase enthalpy of formation Identification and DSC measurements with Cp ones.
  English related to this part seems to be in accordance with the level expected by the journal. So, only a minor change is requested for this part. I suggest adding the experimental temperature ranges used for measurements whose fitted equations are reported in (1) and (2).   Conversely, English in other parts of the manuscript is not suitable for the level recommended by the journal: 
  • sometimes the article "the" is missed;
  • "in function of" rather than the correct "as a function of"
  • line 141: please replace "rate mass loss" with "mass loss rate"
Some relevant critical issues must be fixed in the revision of the paper. 1) How can the authors demonstrate that evaporation took place without appreciable a simultaneous decomposition? 2) Why did the authors not take into account a possible influence of diffusion on the rate of mass loss, as indicated in a previous study (Ref. 51)? In my opinion, diffusion phenomena may affect the rates of mass loss and consequently the sublimation enthalpies so determined. So, the authors must check the effective potential influence and explain clearly and briefly in the text why they consider it negligible. 3) A wide experimental temperature range explored (100 K) seems to me too high to consider the reaction enthalpy change substantially constant. Actually a small (but not negligible) decrease of the slopes in Fig. 2 is observed with increasing the temperature (going from the right to the left side of the plot), although the r2 coefficient is very close to unity.  A comment about this behavior is needed by the authors.   In addition, there are some minor changes to make during the revision of the manuscript. 4) Please, replace "fusion temperature" with the correct "melting temperature". 5) Table 7: the structural formulas of the products in the reaction (R7.2) are not well reported. Please, revise it. 6) lines 367-368: there is no need to report the range of heating rates covered by the apparatus but only those adopted in the experiments.   Due to the necessity to explain and solve the critical points (1-3) I believe the authors need more time, eventually to carry out further measurements to exclude (if so) the potential influence of diffusion phenomena in the TG evaluation of the sublimation enthalpy, although many parts of the manuscript are rigorous, well written and clearly presented.   So, my decision is for major revision.

Author Response

Dear reviewer, I hope you are very well.

I am only writing to comment on the following.

Thank you very much for the comments you made about the manuscript because I know they will help improve it.

I am attaching a file with the answers to the questions generated.
Hoping to have a favourable response from you, I remain at your service.

Sincerely

Arthur

Reviewer 2 Report

The authors have chosen to study two highly appropriate, interesting and important molecules,tetra-N-phenylbenzidine and 3,3′-bis (N-carbazolyl)-1,1′-biphenyl.  I have two questions and "judgment calls". The first is why the authors did not measure, or even discuss the corresponding 3,3’-bis(diphenylamino)biphenyl and 4,4′-bis (N-carbazolyl)-1,1′-biphenyl. I looked more closely and found ref. 33 discussed the latter species I was asking about as well as t tetra-N-phenylbenzidine of current interest. I also note there is uncited literature thermochemical study on N-phenylcarbazole, Emel'yanenko, V. N.; Zaitsau, D. H.; Pimerzin, A. A.; Verevkin, S. P., J. Chem. Thermodyn (2019), 132, 122-128. Authors, is there any reason why any of these studies were ignored? Or should these comparisons await a later paper?

Author Response

(The authors gave the same response as above.)

Round 2

Reviewer 1 Report

The revised version of the manuscript included all the changes suggested in the previous review and explanation of a couple of obscure issues were provided either in the reply to the Reviewer Report and in the text of the manuscript. Clarity and scientific coundness seems to me significantly improved in the TG experimental part. English style seems to be improved.

I can recommend the publication of the manuscript in the current form.